# Minimum Divergence vs. Maximum Margin: an Empirical Comparison on Seq2Seq Models

**Huan Zhang, Hai Zhao** [*]
Department of Computer Science and Engineering
Shanghai Jiao Tong University
zhanghuan0468@gmail.com, zhaohai@cs.sjtu.edu.cn

## Abstract

Sequence to sequence (seq2seq) models have become a popular framework for neural sequence prediction. While traditional seq2seq models are trained by Maximum Likelihood Estimation (MLE), much recent work has made various attempts to optimize evaluation scores directly to solve the mismatch between training and evaluation, since model predictions are usually evaluated by a task specific evaluation metric like BLEU or ROUGE scores instead of perplexity. This paper puts this existing work into two categories, a) minimum divergence, and b) maximum margin. We introduce a new training criterion based on the analysis of existing work, and empirically compare models in the two categories. Our experimental results show that our new training criterion can usually work better than existing methods, on both the tasks of machine translation and sentence summarization.

## 1 Introduction

Sequence to sequence (seq2seq) models (Kalchbrenner & Blunsom, 2013; Sutskever et al., 2014; Bahdanau et al., 2015; Luong et al., 2015) are a powerful end-to-end solution to sequence prediction since they allow a mapping between two sequences of different lengths. However, Maximum Likelihood Estimation (MLE) which is used as a standard training criterion in seq2seq has a series of drawbacks:

- Training and evaluation mismatch: during training, we maximize the log likelihood, while during inference, the model is evaluated by a different metric such as BLEU or ROUGE;
- MLE fails to assign proper scores to different model outputs, which means that all incorrect outputs are treated equally.

Similar to SEARN (Daumé et al., 2009) and DAGGER (Ross et al., 2011) in traditional models, many reinforcement learning (RL) techniques have been extended to seq2eq models recently to solve the mismatch between training and inference. The key idea is to replace MLE with a criterion that gives task-specific scores to different model outputs.

Recent approaches to taking the evaluation metric into training criteria generally fall into one of the following two categories:

1. *minimum divergence*: Minimum divergence methods minimize some distance functions between the model output distribution and the probability distribution given by the ground truth, and by optimizing the divergence between the two distributions, the model output distribution will ideally be equal to the ground truth distribution.

2. *maximum margin*: During inference, in structured prediction, the goal is to simply ensure good structures have the highest predicted score. That is, margin based learning differs from *minimum divergence* learning in the aspect that it only requires that the predicted score of the correct output should be higher than that of the others by a margin.

---

[*]Corresponding author.

The main contribution of this paper can be summarized as: based on the analysis of existing work, we introduce a novel training criterion for seq2seq learning and show its effectiveness.

Based on our experimental results on the tasks of machine translation and sentence summarization, we conclude that some models that take the evaluation metric into consideration can improve over a strong MLE baseline by a large margin, and our new training criterion can usually get better results than existing work.

## 2 RELATED WORK

Seq2seq models were proposed and improved by Kalchbrenner & Blunsom (2013) and Sutskever et al. (2014). With the attention mechanism (Bahdanau et al., 2015; Luong et al., 2015), seq2seq models achieve better results than traditional models in many fields including machine translation (Bahdanau et al., 2015; Luong et al., 2015) and text summarization (Chopra et al., 2016).

Before seq2seq models, previous work on optimizing task specific evaluation metric scores generally includes SEARN (Daumé et al., 2009), DAGGER (Ross et al., 2011), Minimum Error Rate Training (MERT) (Och, 2003), and softmax-margin criterion (Gimpel & Smith, 2010).

Seq2seq model uses MLE as the training objective, which also has the problem of training and evaluation mismatch. Ranzato et al. (2016) incorporated the evaluation metric into training of seq2seq models and proposed the mixed incremental cross entropy REINFORCE (MIXER) training strategy, which is similar to the idea of Minimum Risk Training (MRT) (Smith & Eisner, 2006; Li & Eisner, 2009; Ayana et al., 2016; Shen et al., 2016). MIXER used decoder hidden states to predict the bias term to reduce the variance, while MRT renormalized the predicted probabilities.

Inspired by MIXER, Bahdanau et al. (2017) extended the actor critic algorithm (Konda & Tsitsiklis, 2000; Sutton et al., 2000) to the seq2seq framework. Sokolov et al. (2016) and Kreutzer et al. (2017) proposed the pairwise preference learning framework, which enables the model to learn from the difference between a pair of samples generated by the model. Another popular model in the form of *divergence minimization* is the smoothed version of MERT algorithm proposed by Och (2003). The smoothed MERT algorithm also minimizes the expected loss over samples from model outputs and is similar to MRT essentially. Norouzi et al. (2016) proposed Reward Augmented Maximum Likelihood (RAML) which also optimizes the divergence by data augmentation, and RAML was further combined with maximum likelihood via $\alpha$-divergence recently (Koyamada et al., 2017).

In contrast to the above *divergence minimization* based approaches, Wiseman & Rush (2016) used a margin based training objective and their search optimization process requires performing parameter updates or inference each time a new word is generated. One main drawback of *maximum margin* is that given a sample subset of an arbitrary size, only two samples of the subset can contribute to the gradient. The softmax-margin criterion proposed by Gimpel & Smith (2010) can alleviate this problem, in which all samples are used in loss and gradient computation. However, Edunov et al. (2018) shows that the softmax-margin criterion still cannot reach the performance of risk minimization. As the latest work, Edunov et al. (2018) is very similar to ours. However, our interests are in finding better training criteria on a consistent setting, while they mainly focused on absolute performance improvement, using techniques like combination with token level loss, online vs. offline candidate generation, etc.

## 3 SEQUENCE TO SEQUENCE MODEL

This paper considers the seq2seq model from Bahdanau et al. (2015) and Luong et al. (2015), which consists of an RNN encoder and an RNN decoder. Due to the gradient vanishing problem, the vanilla RNN cell is replaced with a Gated Recurrent Unit (GRU) (Cho et al., 2014) or Long Short-Term Memory (LSTM) (Hochreiter & Schmidhuber, 1997). A bidirectional RNN is used as encoder, with one RNN reading the source sequence from beginning to end, and the other RNN reading in reversed order. The bidirectional RNN encoder encodes the input sequence and gets a sequence of hidden states by concatenating the hidden states of each word in the two directions, which will be used as attention information in the decoder RNN.

The decoder RNN is initialized by passing the two final hidden states of the encoder to a feed forward network. The mixed information used as input in the first step $m_0$ is initialized by a zero vector. In decoding step $i$, using the input feeding approach described in Luong et al. (2015), the decoder RNN reads the embedding of the word $w_i$ and the mixed information $m_{i-1}$, to generate the hidden state of current step $h_i$:

$$h_i = f_1(concat(w_i, m_{i-1}), h_{i-1}),$$

where $f_1$ is the RNN cell. The decoder then generates the attention context $c_i$ from encoder using $h_i$. Since there are many different attention mechanisms, we omit the details here. After getting $c_i$, the model mixes $c_i$ and $h_i$ together by a feedforward network $f_2$ to generate $m_i$, which is used as the mixed information in the next step:

$$m_i = f_2(c_i, h_i)$$

Finally, $m_i$ is passed to a feedforward network to generate $o_i$, and the softmax layer outputs the probability for each word at step $i$. Suppose the target word in step $i$ is $v'$, the probability of the target word in step $i$ is given by

$$p(y_i = v' \mid x, y_{1:i-1}) = \frac{\exp(o_{iv'})}{\sum_v \exp(o_{iv})}.$$

For the sequence $y = \langle y_1, y_2, ..., y_l \rangle$, the probability $p(y \mid x)$ is

$$p(y \mid x) = p(y_1 \mid x)\, p(y_2 \mid x, y_1)...p(y_l \mid x, y_{1:l-1}).$$

Suppose $y^*$ is the correct output, MLE actually maximizes the log likelihood of $p(y^* \mid x)$:

$$\mathcal{L}_{\mathbf{MLE}} = -\log p(y^* \mid x) = -\sum_{i=1}^{l} \log p(y_i^* \mid x, y_{<i}^*)$$

For a detailed explanation, please refer to Luong et al. (2015). Now we describe the algorithm to construct a subset $\mathbf{S}$ from all possible output sequences which will be used in Section 4 & 5. In seq2seq model, the *n-best* list can only be approximated by performing beam search, as used in Wiseman & Rush (2016), or Monte Carlo (MC) sampling, as used in Ayana et al. (2016) and Shen et al. (2016). While using beam search may produce more accurate results, the time required for generating a candidate set of the same size as MC sampling using beam search is too long. In this paper, we construct the approximated *n-best* list $\mathbf{S}$ using MC sampling proposed in Shen et al. (2016), and for completeness, we describe the sampling approach in Appendix 8.1 in details. Please note that the sampling algorithm does not guarantee the size of the sample subset, since it is inefficient to force the model to generate an exact number of distinct samples using GPU.

## 4    MINIMUM DIVERGENCE

In this section, we consider training criteria in the form of the following equation:

$$\mathcal{L} = D(p_1 \,\|\, p_2),$$

where $p_1$ and $p_2$ are two probability distributions given by the model or the ground truth, and $D$ is a distance function, for example, Kullback Leibler (KL) divergence or cross entropy. The baseline training criterion MLE can also be seen as minimum divergence training, where the cross entropy between the model output distribution and the one-hot encoded ground truth is minimized. Now we will discuss training criteria in which the ground truth distribution contains information of evaluation metric scores.

### 4.1    REWARD AUGMENTED MAXIMUM LIKELIHOOD

Suppose that $\{x, y^*\}$ is a given input-output pair, the model learns parameters $\theta$ and gives prediction $p(y \mid x; \theta)$ on the output space $\mathbf{Y}$, the *exponentiated payoff distribution* (Norouzi et al., 2016) $q(y \mid y^*)$ is defined as follows:

$$q(y \mid y^*) = \frac{1}{Z(y^*)} \exp[\, r(y, y^*)/\tau\,], \tag{1}$$

where $Z(\boldsymbol{y}^*) = \sum_{\boldsymbol{y} \in \mathbf{Y}} \exp[\, r(\boldsymbol{y}, \boldsymbol{y}^*)/\tau \,]$ is the normalization factor. $\tau$ is a hyperparameter that controls the smoothness of the distribution, and $r(\boldsymbol{y}, \boldsymbol{y}^*)$ is the score of prediction $\boldsymbol{y}$ given by the evaluation metric, e.g. BLEU (Papineni et al., 2002), or ROUGE (Lin, 2004)

The training objective of Reward augmented Maximum Likelihood (RAML) (Norouzi et al., 2016) is defined as

$$\mathcal{L}_{\mathbf{RAML}} = -\mathbb{E}_{q(\boldsymbol{y} \,|\, \boldsymbol{y}^*)}[\, \log p(\boldsymbol{y} \,|\, \boldsymbol{x}; \boldsymbol{\theta}) \,], \tag{2}$$

in which, the gradient is:

$$\mathbb{E}_{q(\boldsymbol{y} \,|\, \boldsymbol{y}^*)}[\, \nabla_{\boldsymbol{\theta}} \log p(\boldsymbol{y} \,|\, \boldsymbol{x}; \boldsymbol{\theta}) \,] = -\sum_{\boldsymbol{y} \in \mathbf{Y}} q(\boldsymbol{y} \,|\, \boldsymbol{y}^*) \, \nabla_{\boldsymbol{\theta}} \log p(\boldsymbol{y} \,|\, \boldsymbol{x}; \boldsymbol{\theta}) \tag{3}$$

Equation (3) requires computing $q(\boldsymbol{y} \,|\, \boldsymbol{y}^*)$ for all $\boldsymbol{y}$ and performing gradient descent on the whole $\mathbf{Y}$. However, due to the huge search space of $\mathbf{Y}$ and the high computational cost of $r(\boldsymbol{y}, \boldsymbol{y}^*)$, Equation (3) cannot be directly computed in an efficient way. Norouzi et al. (2016) proposed using stratified sampling. One can sample from negative edit distance or Hamming distance and then do importance reweighting to reweight the samples with task specific scores. After sampling from $q(\boldsymbol{y} \,|\, \boldsymbol{y}^*)$, the model is trained to maximize $\log p(\boldsymbol{y} \,|\, \boldsymbol{x}; \boldsymbol{\theta})$.

Another choice is to construct a sample subset $\mathbf{S}$ from the model outputs using MC sampling, and use Equation (1) to compute the probability $q'(\boldsymbol{y} \,|\, \boldsymbol{y}^*)$ in the sample subset. The training criterion is then changed to

$$\mathcal{L}'_{\mathbf{RAML}} = -\sum_{\boldsymbol{y} \in \mathbf{S}} q'(\boldsymbol{y} \,|\, \boldsymbol{y}^*) \, \log p(\boldsymbol{y} \,|\, \boldsymbol{x}; \boldsymbol{\theta}),$$

where

$$q'(\boldsymbol{y} \,|\, \boldsymbol{y}^*) = \frac{\exp[\, r(\boldsymbol{y}, \boldsymbol{y}^*)/\tau \,]}{\sum_{\boldsymbol{y}' \in \mathbf{S}} \exp[\, r(\boldsymbol{y}', \boldsymbol{y}^*)/\tau \,]}.$$

## 4.2 Minimum Risk Training

We first introduce the REINFORCE algorithm (Williams, 1992), which is a famous policy gradient method in RL, and then compare it with the Minimum Risk Training (MRT) (Smith & Eisner, 2006; Li & Eisner, 2009; Ayana et al., 2016; Shen et al., 2016). In REINFORCE, given a state $\boldsymbol{x}$, an action $\boldsymbol{y}$ under state $\boldsymbol{x}$, and model parameters $\boldsymbol{\theta}$, the policy gradient $\boldsymbol{g}$ is

$$\boldsymbol{g} = (r(\boldsymbol{y}) - \mathcal{B}(\boldsymbol{x}))\nabla_{\boldsymbol{\theta}} \log p(\boldsymbol{y} \,|\, \boldsymbol{x}; \boldsymbol{\theta}), \tag{4}$$

where $r(\boldsymbol{y})$ is the reward of action $\boldsymbol{y}$, and $\mathcal{B}(\boldsymbol{x})$ is the *baseline* or *control variate* that is used to reduce the variance (without introducing bias).

In MRT, the training objective is the expectation of evaluation metric scores taken with respect to the model output distribution:

$$\mathcal{L}_{\mathbf{MRT}} = -\mathbb{E}_{p(\boldsymbol{y} \,|\, \boldsymbol{x}; \boldsymbol{\theta})}[\, r(\boldsymbol{y}, \boldsymbol{y}^*) \,]. \tag{5}$$

Also due to the huge search space, Equation (5) can only be estimated by sampling from the model outputs. Suppose $\mathbf{S}$ is a sample subset from $\mathbf{Y}$, $p(\boldsymbol{y} \,|\, \boldsymbol{x}; \boldsymbol{\theta})$ is replaced by $p'(\boldsymbol{y} \,|\, \boldsymbol{x}; \boldsymbol{\theta}; \alpha)$:

$$p'(\boldsymbol{y} \,|\, \boldsymbol{x}; \boldsymbol{\theta}; \alpha) = \frac{p(\boldsymbol{y} \,|\, \boldsymbol{x}; \boldsymbol{\theta})^{\alpha}}{\sum_{\boldsymbol{y}' \in \mathbf{S}} p(\boldsymbol{y}' \,|\, \boldsymbol{x}; \boldsymbol{\theta})^{\alpha}}, \tag{6}$$

where $\alpha$ is a hyperparameter that controls the smoothness of $p'(\boldsymbol{y} \,|\, \boldsymbol{x}; \boldsymbol{\theta}; \alpha)$. We will later compare the effect of $\alpha$ here with $\tau$ in RAML.

The gradient of $\mathcal{L}_{\mathbf{MRT}}$ can be derived as:

$$-\nabla_{\boldsymbol{\theta}} \, \mathbb{E}_{p'(\boldsymbol{y} \,|\, \boldsymbol{x}; \boldsymbol{\theta}, \alpha)}[\, r(\boldsymbol{y}, \boldsymbol{y}^*) \,] = -\alpha \, \mathbb{E}_{p'(\boldsymbol{y} \,|\, \boldsymbol{x}; \boldsymbol{\theta}, \alpha)}[\, \nabla_{\boldsymbol{\theta}} \log p(\boldsymbol{y} \,|\, \boldsymbol{x}; \boldsymbol{\theta})(r(\boldsymbol{y}, \boldsymbol{y}^*) - B) \,],$$

with

$$B = \mathbb{E}_{p'(\boldsymbol{y}' \,|\, \boldsymbol{x}; \boldsymbol{\theta}, \alpha)}[\, r(\boldsymbol{y}', \boldsymbol{y}^*) \,]. \tag{7}$$

If we regard the whole seq2seq framework as a one-step Markov Decision Process (MDP), in which the input sequence $\boldsymbol{x}$ is the state, the output sequence $\boldsymbol{y}$ is the action, and the evaluation metric score

$r$ is the reward, both REINFORCE and MRT actually optimize the same training criterion. The difference between them is that REINFORCE requires only a single sample and is unbiased, while MRT has multiple samples and the probability renormalization in Equation (6) introduces bias. The *expected loss* objective in Kreutzer et al. (2017) is also similar (and closer) to REINFORCE which uses one sample to estimate the gradient.

### 4.3   COMPARISON OF RAML & MRT

The cross entropy is defined as:

$$D_{\textbf{XENT}}(p_1(x)||p_2(x)) = -\mathbb{E}_{p_1(x)}[\log p_2(x)]. \tag{8}$$

Starting from RAML, in terms of cross entropy, Equation (2) can be rewritten as:

$$\mathcal{L}_{\textbf{RAML}} = D_{\textbf{XENT}}(q(\boldsymbol{y}\,|\,\boldsymbol{y^*})\,||\,p(\boldsymbol{y}\,|\,\boldsymbol{x};\boldsymbol{\theta}))$$

The training criterion of RAML is essentially minimizing the the cross entropy loss from $q(\boldsymbol{y}\,|\,\boldsymbol{y^*})$ to $p(\boldsymbol{y}\,|\,\boldsymbol{x};\boldsymbol{\theta})$.

Meanwhile, the training criterion of MRT in Equation (5) can be rewritten as:

$$
\begin{aligned}
\mathcal{L}_{\textbf{MRT}} &= -\tau\,\mathbb{E}_{p(\boldsymbol{y}\,|\,\boldsymbol{x};\boldsymbol{\theta})}[\,r(\boldsymbol{y},\boldsymbol{y^*})/\tau\,] \\
&= -\tau\,\mathbb{E}_{p(\boldsymbol{y}\,|\,\boldsymbol{x};\boldsymbol{\theta})}[\,\log q(\boldsymbol{y}\,|\,\boldsymbol{y^*})\,] - \tau\,\mathbb{E}_{p(\boldsymbol{y}\,|\,\boldsymbol{x};\boldsymbol{\theta})}[\,\log Z(\boldsymbol{y^*})\,] \\
&= -\tau\,\mathbb{E}_{p(\boldsymbol{y}\,|\,\boldsymbol{x};\boldsymbol{\theta})}[\,\log q(\boldsymbol{y}\,|\,\boldsymbol{y^*})\,] - \tau\,\log Z(\boldsymbol{y^*}),
\end{aligned}
$$

in which, the second term is a constant as it has no gradient with respect to $\boldsymbol{\theta}$. Thus we have:

$$\mathcal{L}_{\textbf{MRT}} = \tau\,D_{\textbf{XENT}}(p(\boldsymbol{y}\,|\,\boldsymbol{x};\boldsymbol{\theta})||q(\boldsymbol{y}\,|\,\boldsymbol{y^*})) + Const. \tag{9}$$

Equation (9) actually shows that MRT minimizes the cross entropy loss from $p(\boldsymbol{y}\,|\,\boldsymbol{x};\boldsymbol{\theta})$ to $q(\boldsymbol{y}\,|\,\boldsymbol{y^*})$.

By comparing the above two methods, we find that both RAML and MRT are minimizing the cross entropy loss between the model output distribution and the *exponentiated payoff distribution*, but with different directions of $D_{\textbf{XENT}}$. Now we will discuss the effect of hyperparameter $\tau$ in RAML and $\alpha$ in MRT.

Cross entropy loss has a property that it is minimized if and only if the two probability distributions are equal. Ideally, when the cross entropy loss in RAML is minimized, we have:

$$
\begin{aligned}
p(\boldsymbol{y}\,|\,\boldsymbol{x};\boldsymbol{\theta}) &= q(\boldsymbol{y}\,|\,\boldsymbol{y^*}) \\
\log p(\boldsymbol{y}\,|\,\boldsymbol{x};\boldsymbol{\theta}) &= r(\boldsymbol{y},\boldsymbol{y^*})/\tau + Const \\
r(\boldsymbol{y},\boldsymbol{y^*}) &= \tau\,\log p(\boldsymbol{y}\,|\,\boldsymbol{x};\boldsymbol{\theta}) + Const.
\end{aligned}
$$

And in MRT [1],

$$\mathcal{L}_{\textbf{MRT}} = -\mathbb{E}_{p'(\boldsymbol{y}\,|\,\boldsymbol{x};\boldsymbol{\theta})}\Big[\log\frac{\exp r(\boldsymbol{y},\boldsymbol{y^*})}{\sum_{\boldsymbol{y'}}\exp r(\boldsymbol{y'},\boldsymbol{y^*})}\Big] + Const.$$

When the two probability distributions are equal, we have:

$$r(\boldsymbol{y},\boldsymbol{y^*}) = \alpha\,\log p(\boldsymbol{y}\,|\,\boldsymbol{x};\boldsymbol{\theta}) + Const. \tag{10}$$

During training of both RAML and MRT, the model is trying to predict the evaluation score $r$ by multiplying $\log p$ with a constant (either $\tau$ in RAML or $\alpha$ in MRT).

### 4.4   HELLINGER DISTANCE MINIMIZATION

Now we introduce another popular distance metric, the Hellinger distance (Hellinger, 1909), which also belongs to the family of Csiszár f-divergence (Csiszár, 1963) and has been used in graphical models (Beykikhoshk et al., 2015) and clustering algorithms (Abdel-Azim, 2016; Ji et al., 2018).

---

[1] Here we refer to the MRT model in Shen et al. (2016), which contains sampling and renormalization as described in Section 4.2.

Given two probability distributions $p_1$ and $p_2$, let $f$ and $g$ denote the probability density functions of $p_1$ and $p_2$. The square of the Hellinger distance $H(p_1, p_2)$ is defined as:

$$H^2(p_1, p_2) = \frac{1}{2} \int (\sqrt{f(x)} - \sqrt{g(x)})^2 \, dx.$$

The training criterion using squared Hellinger distance is:

$$\mathcal{L}_{\mathbf{Hellinger}} = \sum_{\boldsymbol{y} \in \mathbf{S}} (\sqrt{p'(\boldsymbol{y} \mid \boldsymbol{x}; \boldsymbol{\theta})} - \sqrt{q(\boldsymbol{y} \mid \boldsymbol{y^*})})^2,$$

with $p'(\boldsymbol{y} \mid \boldsymbol{x}; \boldsymbol{\theta})$ defined in Equation (6) and $q(\boldsymbol{y} \mid \boldsymbol{y^*})$ defined in Equation (1).

## 5 MAXIMUM MARGIN

Let $\Delta$ denote the loss given by the evaluation metric:

$$\Delta(\boldsymbol{y}, \boldsymbol{y^*}) = r(\boldsymbol{y^*}, \boldsymbol{y^*}) - r(\boldsymbol{y}, \boldsymbol{y^*}).$$

$\Delta$ is 0 when $\boldsymbol{y} = \boldsymbol{y^*}$ and positive otherwise. Let $F$ be the predicted score of the model:

$$F = \tau \, \log p(\boldsymbol{y} \mid \boldsymbol{x}; \boldsymbol{\theta}).$$

Different from minimizing the expected risk, another choice of training criterion is to optimize the evaluation score of the sequence with the highest predicted score:

$$\arg\min_{\boldsymbol{\theta}} \Delta(\bar{\boldsymbol{y}}, \boldsymbol{y^*}), \tag{11}$$

where

$$\bar{\boldsymbol{y}} = \arg\max_{\boldsymbol{y}} F(\boldsymbol{y}, \boldsymbol{\theta}).$$

For each input sequence $\boldsymbol{x}$, we first do inference to find the output $\bar{\boldsymbol{y}}$ that has the highest predicted score, then we train the model to maximize the score of $\bar{\boldsymbol{y}}$ given by evaluation metric.

However, directly optimizing the training criterion that includes an $\arg\max$ is not computationally tractable. In structured prediction, this problem has been widely discussed in the training process of Structural Support Vector Machine (SSVM) (Tsochantaridis et al., 2004; Joachims, 2006; Joachims et al., 2009). Tsochantaridis et al. (2004) proposed using margin and slack rescaling instead. Margin Rescaling is based on the following inequality:

$$\Delta(\bar{\boldsymbol{y}}, \boldsymbol{y^*}) \le \Delta(\bar{\boldsymbol{y}}, \boldsymbol{y^*}) + F(\bar{\boldsymbol{y}}, \boldsymbol{\theta}) - F(\boldsymbol{y^*}, \boldsymbol{\theta}) \le \max_{\boldsymbol{y}}(\Delta(\boldsymbol{y}, \boldsymbol{y^*}) + F(\boldsymbol{y}, \boldsymbol{\theta})) - F(\boldsymbol{y^*}, \boldsymbol{\theta}) \tag{12}$$

Similarly, Slack Rescaling is:

$$\Delta(\bar{\boldsymbol{y}}, \boldsymbol{y^*}) \le \Delta(\bar{\boldsymbol{y}}, \boldsymbol{y^*})(1 + F(\bar{\boldsymbol{y}}, \boldsymbol{\theta}) - F(\boldsymbol{y^*}, \boldsymbol{\theta})) \le \max_{\boldsymbol{y}} \Delta(\boldsymbol{y}, \boldsymbol{y^*})(1 + F(\boldsymbol{y}, \boldsymbol{\theta}) - F(\boldsymbol{y^*}, \boldsymbol{\theta})) \tag{13}$$

As Equation (13) is a tighter bound on $\Delta(\bar{\boldsymbol{y}}, \boldsymbol{y^*})$, slack rescaling generally gives better results than margin rescaling. Both Equation (12) and Equation (13) can be optimized by subgradient descent. Let $\hat{\boldsymbol{y}}$ denote the sample that gives the $\max$ value. The subgradient for margin rescaling is:

$$\nabla_{\boldsymbol{\theta}} \tau \log p(\hat{\boldsymbol{y}}) - \nabla_{\boldsymbol{\theta}} \tau \log(p(\boldsymbol{y^*})),$$

and similarly, the subgradient for slack rescaling is:

$$\Delta(\hat{\boldsymbol{y}}, \boldsymbol{y^*})(\nabla_{\boldsymbol{\theta}} \tau \log p(\hat{\boldsymbol{y}}) - \nabla_{\boldsymbol{\theta}} \tau \log p(\boldsymbol{y^*})).$$

Since doing *loss augmented inference* on such a huge search space is again not realistic, approximate inference is still needed. Equation (12) also has the name of structured hinge loss when used in SSVM. One may have found that Equation (13) is the same as the training criterion in Wiseman & Rush (2016). Besides, to make a fair comparison with other training criteria and make it applicable on large datasets, we replace the beam search with sample subset.

## 6 EXPERIMENTS

We evaluate the training criteria on two tasks in total. The first task is machine translation on a small dataset to explore the hyperparameter impact. The second task is sentence summarization using large datasets.

For all the experiments, we use LSTM as RNN cell. The attention mechanism used in our experiments is global attention with input feeding (Luong et al., 2015).

### 6.1 MACHINE TRANSLATION

Due to space limit, experimental results on hyperparameter effects and training details are provided in Appendix 8.2.1.

**Data** We use the IWSLT 2014 German-English translation dataset, with the same splits as Ranzato et al. (2016) and Wiseman & Rush (2016), which contains about 153K training sentence pairs, 7K validation sentence pairs and 7K test sentence pairs. Sentences in the dataset are first tokenized and then converted into lowercase. During training, the maximum length for both inputs and outputs is restricted to 50. Words that appear less than three times are replaced with a UNK token.

**Models** The encoder is a single-layer bidirectional LSTM with 256 hidden units for either direction, and the decoder LSTM also has 256 hidden units. The size of word embedding for both encoder and decoder is 256. We use a dropout rate (Srivastava et al., 2014) of 0.2 to avoid overfitting. The gradient is clipped when its norm exceeds 1.0 to prevent gradient explosion and stabilize learning (Pascanu et al., 2013). The batch size is set to 32 and the training set is shuffled at each new epoch. All models are trained with the Adam optimizer (Kingma & Ba, 2015). The sentence level BLEU score used as reward or loss during training is smoothed by adding 1 to both numerator and denominator, as suggested by Chen & Cherry (2014).

**Results** For the two max margin methods, $\tau$ is set to $1.0 \times 10^{-3}$. We use the sample subset version of RAML, and $\tau$ for RAML is also set to $1.0 \times 10^{-3}$. For Hellinger loss, $\alpha = 5.0 \times 10^{-4}$ and $\tau = 0.5$. The final results on the test set are described in Table 1. Max margin methods here improve the MLE baseline less significantly than Wiseman & Rush (2016) did, since the search optimization algorithm is not used in our experiments. The training criterion of squared Hellinger distance gives the highest BLEU score on the test set.

Table 1: BLEU scores on IWSLT German-English translation evaluated on the test set.

| criterion | greedy | beam |
|---|---|---|
| MLE | 26.03 | 27.17 |
| margin | 26.22 | 27.32 |
| slack | 26.32 | 27.37 |
| RAML | 26.34 | 27.36 |
| MRT | 27.35 | 28.10 |
| Hellinger | **27.75** | **28.38** |

### 6.2 SENTENCE SUMMARIZATION

The task of sentence summarization is: given a long sentence or a passage, the model will generate a short sentence that summarizes the long text. For evaluation, following Rush et al. (2015), we use the full length F1 variant of ROUGE score (Lin, 2004) as metric. Training details are described in Appendix 8.2.2.

**Data** We use the Gigaword corpus with the same preprocessing steps as in Rush et al. (2015). In the Gigaword dataset, the first sentence of each article is regarded as the source sequence, and the title of the article as the target sequence. The training set consists of 3.7M sequence pairs, and the vocabulary size for both article and title is 50K, with the out-of-vocabulary words replaced by a unified UNK token. The whole dataset is then converted into lowercase without tokenization.

During training, we use the first 2K sequences of the dev corpus as validation set, and the size of the test set is also 2K. The maximum length for both input and output sequence is restricted to 50.

**Models** The encoder is a single layer bidirectional LSTM with 1000 hidden units for either direction. The single layer decoder LSTM also has 1000 dimensions. The word embedding size for both encoder and decoder is 500. Dropout is not used in the sentence summarization task, and the gradient is clipped when its norm exceeds 1.0. The batch size is set to 128. The training set is shuffled at each new epoch, and we prefetch 20 batches and sort them by the length of target sequence to form new batches to speed up training. We use ROUGE-2 as the reward or loss function.

**Results** Table 2 shows the results on the validation set and the test set. We report ROUGE-1, ROUGE-2, ROUGE-L and ROUGE-S4 score. MRT outperforms the MLE baseline by a large margin, but is still slightly lower than the newly introduced Hellinger loss.

Table 2: Results on Gigaword sentence summarization task

| criterion | ROUGE-1 | | ROUGE-2 | | ROUGE-L | | ROUGE-S4 | |
|---|---|---|---|---|---|---|---|---|
| | valid | test | valid | test | valid | test | valid | test |
| MLE | 48.41 | 34.20 | 24.14 | 15.22 | 45.08 | 31.84 | 22.62 | 14.00 |
| margin | 48.43 | 34.20 | 24.16 | 15.24 | 45.09 | 31.85 | 22.65 | 14.00 |
| slack | 48.40 | 34.20 | 24.12 | 15.25 | 45.08 | 31.83 | 22.63 | 14.00 |
| RAML | 48.43 | 34.20 | 24.16 | 15.24 | 45.10 | 31.84 | 22.66 | 14.00 |
| MRT | 50.41 | 35.63 | 26.42 | 16.96 | 47.05 | 33.24 | 24.47 | 15.31 |
| Hellinger | **51.02** | **35.84** | **26.88** | **17.03** | **47.71** | **33.47** | **25.12** | **15.54** |
| MLE + BS | 49.69 | 36.21 | 25.52 | 16.37 | 46.55 | 32.75 | 24.00 | 15.01 |
| MRT + BS | 50.72 | 36.47 | 26.74 | **17.62** | 47.41 | **34.05** | 25.05 | 15.98 |
| Hellinger + BS | **51.07** | **36.59** | **26.95** | 17.61 | **47.62** | 34.03 | **25.34** | **16.04** |

## 7 DISCUSSION

For training criteria in the form of *divergence minimization*, according to our experimental results, MRT outperforms RAML. As mentioned before, MRT and RAML minimize the cross entropy in two different directions. By the definition of cross entropy loss in Equation (8), it is only minimized when $p_1 = p_2$. Optimizing the cross entropy in either of the two directions both guarantee consistency, i.e., the training procedure can ultimately recover the true probability distribution.

For a model that minimizes $D_{\mathbf{XENT}}(p_1||p_2)$, the model will sample from $p_1$ and maximize $\log p_2$, which means that if $x$ has a high probability in $p_1$, it will also have a high probability in $p_2$, but if has a low probability in $p_1$, $p_2(x)$ may still be high.

Thus in RAML which samples from *exponentiated payoff distribution* $q$ and maximizes $\log p$, sequences that have a high evaluation score will also have a high predicted probability. However, samples that have a low $q$, e.g., a low BLEU or ROUGE, may still have a high $p$.

In contrast, MRT will first sample from model outputs and ensure that a high BLEU or ROUGE score will be assigned to sequences that have a high probability in model output distribution $p$, at the cost that sequences that have low probabilities in $p$ may still have high evaluation scores. This can explain the difference between RAML and MRT.

The squared Hellinger distance can be regarded as the squared error between $p^{\frac{1}{2}}$ and $q^{\frac{1}{2}}$, which is essentially a regression loss, while the cross entropy loss usually works better in classification problems. By optimizing the cross entropy loss, the model is trained to find the candidate that has the highest probability and pays less attention to samples that have low probabilities, while by optimizing the squared Hellinger distance, the model learns to predict the evaluation metric score for every sample in the candidate set and pays equal attention to all of them. Since seq2seq models

generate a predicted sequence word by word, the target sequence which has the highest evaluation metric score (sample from $q$ and maximize $\log p$) or the highest predicted probability (sample from $p$ and maximize $\log q$) may not be reached by approximate inference algorithms like greedy search or beam search. In this case, a model trained by optimizing the cross entropy loss may not work as expected, however, a model trained by optimizing the squared Hellinger distance can still make predictions normally since it pays equal attention to samples with low probabilities during training.

For *maximum margin* methods, slack rescaling is a tighter bound and the experimental result is slightly higher than margin rescaling on our small scale experiments. However, slack rescaling still cannot achieve the same level of performance of MRT and Hellinger distance. The reason may be that the upper bound is still not tight enough. In the training process of *maximum margin* methods, only the target sequence and a negative sample selected from the candidate set are used to update model parameters, which may also explain their performance.

### ACKNOWLEDGMENTS

This paper was partially supported by National Key Research and Development Program of China (No. 2017YFB0304100), Key Projects of National Natural Science Foundation of China (U1836222 and 61733011), Key Project of National Society Science Foundation of China (No. 15-ZDA041), The Art and Science Interdisciplinary Funds of Shanghai Jiao Tong University (No. 14JCRZ04).

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

## 8 APPENDIX

### 8.1 THE SAMPLING ALGORITHM

For completeness, we describe the MC sampling algorithm proposed by Shen et al. (2016) and Ayana et al. (2016) here.

---

**Algorithm 1** The sampling approach to constructing the approximated *n-best* list

---

**Input:**
    source sequence $x$, model parameters $\theta$, target sequence $y^*$
    sample size $m$, maximum length $l$
**Output:**
    A set of samples $\mathbf{S}$ and corresponding probability $\mathbf{P}$
 1: $\mathbf{S} = \{y^*\}$; $\mathbf{P} = \{p(y^*)\}$;
 2: **for** $i$ in $1, ..., m-1$ **do**
 3:     sample a sequence word by word given $x$ and $\theta$, stop sampling when length is $l$ or meeting $\langle end\text{-}of\text{-}sequence\rangle$ symbol, and finally store the result and probability in $y$ and $q$ respectively

 4:     **if** $y$ in $\mathbf{S}$ **then**
 5:         continue
 6:     **end if**
 7:     $\mathbf{S} = \mathbf{S} \cup \{y\}$
 8:     $\mathbf{P} = \mathbf{P} \cup \{q\}$
 9: **end for**
10: **return** $\mathbf{S}, \mathbf{P}$

---

### 8.2 EXPERIMENTS

#### 8.2.1 MACHINE TRANSLATION

The MLE baseline is trained with a learning rate of $3.0 \times 10^{-4}$. The model is trained for 20 epochs in total, and is evaluated every 1000 steps on the validation set.

$\tau$ **for RAML** For RAML, since we use BLEU instead of Hamming distance as evaluation metric, we first show the BLEU scores on the validation set for hyperparameter $\tau$ in Table 3. RAML-I refers to importance reweighting and RAML-S refers to sample subset RAML, as mentioned in Section 4.1. Experimental results show that the BLEU score is not very sensitive to hyperparameter $\tau$ although $\tau$ varies from $1.0 \times 10^{-7}$ to $1.0 \times 10^{-1}$, and setting $\tau$ larger will decrease the BLEU score compared with the MLE baseline. We also find that RAML-S tends to give slightly better results than RAML-I. The learning rate for RAML is $1.0 \times 10^{-6}$.

Table 3: Tokenized BLEU scores on IWSLT 2014 German-English translation evaluated on validation set using RAML criterion.

| $\tau$ | RAML-I | RAML-S |
|---|---|---|
| $1.0 \times 10^{-7}$ | 29.25 | 29.28 |
| $1.0 \times 10^{-5}$ | 29.27 | 29.30 |
| $1.0 \times 10^{-3}$ | 29.26 | 29.32 |
| $1.0 \times 10^{-1}$ | 29.28 | 29.32 |

$\alpha$ **for MRT** For MRT, since Shen et al. (2016) did several experiments to study the effect of hyperparameters, the original hyperparameters are strictly followed in our experiments. Hyperparameter $\alpha$ is set to $5.0 \times 10^{-3}$ and the sample size is 100. The best BLEU score of MRT on the validation set is 30.55. The learning rate is set to $1.0 \times 10^{-5}$.

$\alpha$ **and** $\tau$ **for Hellinger loss** As shown in Table 4, $\alpha = 5.0 \times 10^{-4}$, $\tau = 0.5$ produces the best result. The highest BLEU score on the validation set is 31.13, which is 0.58 higher than MRT. The learning rate used in Hellinger loss is also $1.0 \times 10^{-5}$.

Table 4: Tokenized BLEU scores on IWSLT 2014 German-English translation evaluated on the validation set using squared Hellinger distance with different $\alpha$ and $\tau$ as training criterion.

| $\alpha$ \ $\tau$ | 0.1 | 0.5 | 1.0 |
|---|---|---|---|
| $5.0 \times 10^{-3}$ | 29.33 | 29.71 | 29.22 |
| $5.0 \times 10^{-4}$ | 29.91 | **31.13** | 31.01 |
| $5.0 \times 10^{-5}$ | 30.75 | 30.83 | 31.00 |

**Sample size for Hellinger loss** For squared Hellinger loss, the effect of sample size is shown in Table 5. A larger sample size generally gives a higher BLEU score, however, due to the limit of GPU memory, we cannot set sample size to a larger value.

Table 5: Tokenized BLEU scores on IWSLT 2014 German-English translation evaluated on the validation set using squared Hellinger Distance with different sample sizes.

| sample size | 20 | 50 | 100 |
|---|---|---|---|
| BLEU | 29.51 | 30.30 | **31.13** |

$\tau$ **for max margin** For the two max margin training criteria, the learning rate is $1.0 \times 10^{-7}$. Table 6 shows the effect of $\tau$: according to the results on the validation set, $\tau$ is not very sensitive to the BLEU scores. As shown in Table 6, BLEU scores given by slack rescaling are usually 0.1 higher than for margin rescaling, and we use $\tau = 1.0 \times 10^{-3}$ for both of the two models.

Table 6: Tokenized BLEU scores on IWSLT 2014 German-English translation evaluated on the validation set using max margin criteria.

| $\tau$ | Slack | Margin |
|---|---|---|
| $1.0 \times 10^{-1}$ | 29.16 | 29.23 |
| $1.0 \times 10^{-3}$ | 29.33 | 29.25 |
| $1.0 \times 10^{-5}$ | 29.35 | 29.16 |

### 8.2.2 SENTENCE SUMMARIZATION

**Details** We use ROUGE-2 as reward or loss function. For the MLE baseline, we train the model for 7 epochs in total. For the first 5 epochs, we use Adam with a learning rate of $3.0 \times 10^{-4}$, we then use SGD with a learning rate of $0.1$, and decay the learning rate by a factor of $0.5$ at the beginning of epoch 7.

For the other training criteria, we use the hyperparameters with the best performance described in Section 6.1.

### 8.2.3 ADDITIONAL EXPERIMENTS

**Details** We implement the squared Hellinger distance criterion on convolutional seq2seq model (Gehring et al., 2017) and strictly follow the settings of IWSLT experiments in Edunov et al. (2018).

Table 7: BLEU scores using convolutional seq2seq models.

| criterion | MLE | Hellinger |
|---|---|---|
| BLEU | 32.14 | 32.30 |

