# OpenReview forum: "Minimum Divergence vs. Maximum Margin: an Empirical Comparison on Seq2Seq Models"
_ICLR.cc/2019/Conference_

### Official Review · AnonReviewer2 · 2018-11-02
**Well written and interesting; experiments could be improved**

**Rating:** 7
**Confidence:** 4

**Review:**

In this paper the authors distinguish between two families of training objectives for seq2seq models, namely, divergence minimization objectives and max-margin objectives. They primarily focus on the divergence minimization family, and show that the MRT and RAML objectives can be related to minimizing the KL divergence between the model's distribution over outputs and the "exponentiated payoff distribution," with the two objectives differing in terms of the direction of the KL. In addition, the authors propose an objective using the Hellinger distance rather than the KL divergence, and they conduct experiments on machine translation and summarization comparing all the considered objectives.

The paper is written extremely clearly, and is a pleasure to read. While the discussion of the relationship between RAML and MRT (and MRT and REINFORCE) is interesting and illuminating, many of these insights appear to have been discussed in earlier papers, and the RAML paper itself notes that it differs from REINFORCE style training in terms of the KL direction.

On the other hand, the idea of minimizing Hellinger distance is I believe novel (though related to the alpha-divergence work cited by the authors in the related work section), and it's nice that training with this loss improves over the other losses. Since the authors' results, however, appear to be somewhat below the state of the art, I think the main question left open by the experimental section is whether training with the Hellinger loss would further improve state of the art models. Even if it would not, it would still be interesting to understand why, and so I think the paper could be strengthened either by outperforming state of the art results or, perhaps through an ablation analysis, showing what aspects of current state of the art models make minimizing the Hellinger loss unnecessary.

In summary,

Pros:
- well written and interesting
- a new loss with potential for improvement over other losses
- fairly thorough experiments

Cons:
- much of the analysis is not new
- unclear if the proposed loss will improve the state of the art, and if not why

Update after author response: thanks for your response. I think the latest revision of the paper is improved, and even though state of the art BLEU scores on IWSLT appear to be in the mid 33s, I think the improvement over the Convolutional Seq2seq model is encouraging, and so I'm increasing my score to 7. I hope you'll include these newer results in the paper.

---

> ### Author Response · Authors · 2018-11-22
> **Response**
>
> Thanks for your helpful comments.
>
> 1. “much of the analysis is not new”:
> The relationship between RAML and reinforcement learning based criteria has been discussed in the RAML paper, while our contributions are:
> a) linking MRT to REINFORCE, which is new.
> b) linking the max margin criterion used in [1] to Eq (10), which is new, by deriving from the analysis of RAML and MRT
> c) the training criterion, Hellinger distance, which is new.
>
> We are sorry if you find something already existing in the analysis part, which is sometimes for better background comprehension. For example, it will be quite difficult for readers to understand if we remove the analysis of RAML and MRT.
>
>
>
> 2. comparison with state of the art
> We made a brief survey on previous state of the art models:
>
> 1) Google’s Neural Machine Translation System (no code, no pretrained models)
> 2) Convolutional Sequence to Sequence Learning (code + pretrained models)
> 3) Transformer (code, no pretrained model uses our dataset)
> 4) BERT (code is available, but is still under review now, and extra monolingual dataset is needed)
>
> Considering our familiarity to existing code and the difficulty to modify it, we chose to re-implement our new training criterion on convolutional seq2seq (since BERT is still under review now, and no suggested hyperparameters are provided for Transformer on the small dataset we use). The results on IWSLT dataset are as follows:
>
> criterion                     BLEU
> MLE baseline            32.14
> Hellinger                   32.30
>
> The improvement is quite smaller than the results reported in our paper, however, the improvement does exist even on such a strong baseline. A potential reason for the smaller improvement is the batch size. In the standard implementation of MRT, we use the sample size of 100 and the batch size of 1 (due to the limited number of GPUs we have), while in the README files of both conv-seq2seq and Transformer, the authors stressed the importance of a large batch size.
> Due to the limit of computational resources by our hand, we have no way to explore the impact of the batch size for our case. However, we are aware that the recent reported results from a lot of literature have indicated that the larger batch size plays a crucial role for absolute NMT performance improvement. We are optimistic in hope that the relative small improvement is mostly due to such a factor.
>
> [1] Seq2seq learning as beam search optimization, Wiseman and Rush, EMNLP 2016

---

### Official Review · AnonReviewer1 · 2018-11-03
**some interesting results and connections, but also some technical issues; revisions have improved the paper**

**Rating:** 7
**Confidence:** 4

**Review:**

The authors have updated the paper and clarified some things, and now my impression of the paper has improved. It still feels a little incremental to me, but the potential application areas of these sorts of models are quite large and therefore incremental improvements are not insignificant. This paper suggests some natural follow-up work in exploring Hellinger distance and other variations for these models.

----- original review follows: ------

This paper discusses loss functions for sequence generation tasks that take into account cost functions that reflect the task-specific evaluation metric. They compare RAML and risk (MRT) formally and empirically, and also test a loss based on Hellinger distance. They compare these to some standard max-margin losses. MRT and the Hellinger distance loss perform best in NMT and summarization experiments.

Pros:

There are some interesting aspects of this paper:

- It is interesting to note that RAML and MRT are (similar to) different directions of KL divergences between the same two distributions. (Caveat: the entropy regularizer, which I discuss in "Cons" below.)
- The new Hellinger-distance-based loss seems promising.
- The empirical comparison among losses for standard NMT/summarization tasks is a potentially valuable contribution.

Cons:

A.
The focus/story of the paper need some work. It is unclear what the key contributions are. I think the Hellinger distance loss is potentially the most important contribution, but the authors don't spend much time on that.. it seems that they think the comparison of divergence and max-margin losses is more central. However, I think the authors' conclusion (that the divergence losses are better than the max-margin losses) is not the main story, because RAML is not much better than the max-margin losses. Also, I have some concerns about some of the details of the max-margin losses (listed and discussed below), so I'm not sure how reliable the empirical comparison is.

B.
As for the connections and comparison between RAML and MRT:

It does not seem that MRT corresponds to a divergence of the form given at the start of Sec. 4. There is also an entropy regularizer in Eq. (9). Sec. 4.3 states: "By comparing the above two methods, we find that both RAML and MRT are minimizing the KL divergence between the model output distribution and the exponentiated payoff distribution, but with different directions of D_KL." However, this statement ignores the entropy regularizer in Eq. (9).

Maybe I'm being dense, but I didn't understand where Equation (10) comes from. I understand the equations above it for RAML, but I don't understand the MRT case in Eq. (10). Can you provide more details?

I also don't understand the following sentence: "It turns out that the hyperparameter \tau in RAML and \alpha in MRT have the same effect." What does this mean mathematically? Also, does this equivalence also require ignoring the entropy regularizer? As formulated, L_{RAML} necessarily contains a \tau, but L_{MRT} does not necessarily contain an alpha. It is only when moving to the sample approximation does the alpha become introduced. (MRT does not require this sample approximation; Some older work on MRT developed dynamic programming algorithms to exactly compute the gradients for structured output spaces like sequences, so samples were not used in those cases.) So I think the paper needs to clarify what exactly is meant by the connection between tau and alpha, under what conditions there is a connection between the two, and what exactly is the nature of this connection. If more space is needed for this, many of the details in Sec 3 can be cut or moved to appendices because those are standard and not needed for what follows.

In the experimental results (Sec. 7), MRT outperforms RAML consistently. The authors discuss the impact of the directionality of the KL divergence, but what about the entropy regularizer? It would be interesting to compare MRT both with and without the entropy regularizer. Without the regularizer, MRT would actually correspond to the KL that the authors are describing in the discussion. As it currently stands, two things are changing between MRT and RAML and we don't know which is responsible for the sizable performance gains.

C.
There are several technical issues with the writing (and potentially with the claims/conclusions), most of which are potentially flexible with some corrections and more exposition by the authors:

Is L_{RAML} to be maximized or minimized? Looks like maximized, but clearly both L_{MLE} and L_{MRT} are supposed to be minimized, so the use of L for all of these seems confusing. If different conventions are to be used for each one, it should be explicitly mentioned in each case whether the term is to be maximized or minimized.

At the end of Sec. 4.1, q' is not defined. I can guess what it is, but it is not entirely clear from the context and should be defined.

In Equation 6, please use different notation for the y in the denominator (e.g., y') to avoid collision with the y in the numerator and that on the left-hand side.

The discussion of max-margin losses in Sec. 5 has some things that should be fixed.

1. In Sec. 5, it is unclear why \Delta is defined to be the difference of two r() functions. Why not just make it -r(y, y^*)? Are there some implied conditions on \Delta that are not stated explicitly? If \Delta is assumed to be nonnegative, that should be stated.

2. In Eq. (11), F appears to be a function of y and theta, but in the definition of F, it has no functional arguments. But then further down, F appears to be a function of y only. Please make these consistent.

3. Eq. (11) is not only hard for structured settings; it is also hard in the simplest settings (binary classification with 0-1 loss). This is the motivation for surrogate loss functions in empirical risk minimization for classification settings. The discussion in the paper makes it sound as if these challenges only arise in the structured prediction setting.

4. I'm confused by what the paper is attempting to communicate with Equations 12 and 13. In Eq. 12, y on the left-hand side is not bound to anything, so it is unclear what is being stated exactly. Is it for all y? For any y? In Eq. 13, the \Delta on the right-hand side is outside the max over y -- is that really what was intended? I thought the max (the slack-rescaled loss-augmented inference step) should take into account \Delta. Otherwise, it is just doing an argmax over the score function.

5. If the authors are directly optimizing the right-hand side of the inequality in Equation 12 (as would be suggested for the formula for the gradient), then there is no global minimum of the loss. It would go to negative infinity. Typically people use a "max(0, )" outside the loss so that the global minimum is 0.



Typos and minor issues follow:

Sec. 1:
"the SEARN" --> "SEARN"
"the DAGGER" --> "DAGGER"

Sec. 2:
"genrally" --> "generally"
"took evaluation metric into training" --> "incorporated the evaluation metric into training"
"consistant" --> "consistent"

Sec. 3:
Use \exp instead of exp.
"a detail explanation" --> "a detailed explanation"

Sec. 4.1:
"predition" --> "prediction"

Sec. 4.2:
"only single sample" --> "only a single sample"

Sec. 6.1:
"less significant than" --> "less significantly than"

---

> ### Author Response · Authors · 2018-11-20
> **Response**
>
> Thanks for your comments.
>
> A.
> about the focus/story:
> We’ve changed the writing in abstract and introduction to fit the main story better. Thanks for the helpful comments.
> About the paper structure:
> Actually we did consider changing the paper structure to give more space to Hellinger distance, however, in order to define $p$ and $q$, and explain why we minimize the Hellinger distance between $p$ and $q$, we do think we need to clearly describe RAML and MRT. Many necessary equations and the comparison of existing work have been introduced before the section of Hellinger distance. If we remove the sections of RAML and MRT, it will be more difficult to read.
>
>
> B.
> It would be better to understand if we replace KL divergence by cross entropy, in which case neither RAML nor MRT needs the $q\log q$ term. Now we change the KL divergence to cross entropy.
>
> About Eq 10: We made a mistake here before. $p$ in L_MRT should be replaced by $p’$. (since we are talking about Shen et al.’s MRT for NMT paper) We’ve changed the writing of Eq 10.
>
> About the connection between alpha and tau: we’ve changed the writing here. Our previous thought is only to point out that alpha and tau are in the same place after taking \log and they can be simply understood as smoothing techniques. Since MT is more an engineering problem than a theoretical one, we sometimes cannot expect a perfect link among existing work. Our main goal of introducing the link among previous work (as mentioned by reviewer 2, some similar discussions have appeared in a previous paper) is to explain how our idea of using Hellinger distance comes. We’ve also changed the writing here.
>
> About the regularizer: as mentioned before, if we replace KL by cross entropy, then the comparison of having or not having the regularizer seems unnecessary. We will do some experiments on this question if we have enough time, while our main focus in the past two weeks was to address Reviewer 2’s concern (trying to rebase MRT and Hellinger distance onto a state of the art model).
>
>
> C.
> Both $L$ need to be minimized now. Thanks for pointing it out.
> $q’$ is now defined.
> Eq 6 has been corrected now.
>
>
> Discussion of max margin:
> 1. \Delta needs to be nonnegative (otherwise Eq 13 will be wrong). We’ve changed the writing here.
> 2. This has been corrected now.
> 3. We’ve changed writing here.
> 4. Previously we made some mistakes when writing this part. Now we have stated it more clearly.
> 5. In the new Eq 12, the rightmost part is an upper bound of \Delta. Since \Delta is always nonnegative, the upper bound should also be nonnegative. Thus we don’t think it necessary to add a “max(0,)”.
>
> The typos have been corrected now.
>
> Again, thanks for your helpful comments.

---

> ### Author Response · Authors · 2018-11-25
> **Manuscript Update**
>
> We’ve uploaded a new manuscript to address the following problems:
> 1) We modified the abstract and introduction to highlight our main contributions.
> 2) We analyzed the similarity and difference between RAML and MRT using cross entropy now, which is more precise, since it doesn’t need the entropy regularizer.
> 3) We changed the writing in max margin section and corrected a lot of math problems.
> 4) Other changes included correcting typos, changing wrong notations, etc. And both $L$ need to be minimized now.

---

### Official Review · AnonReviewer3 · 2018-11-05
**This paper compares several well known methods and one new method based on Hellinger distance**

**Rating:** 5
**Confidence:** 4

**Review:**

This paper compares several well known methods and illustrates some connections among these methods, and proposes one new method based on Hellinger distance for seq2seq models' training, experimental results on machine translation and sentence summarization show that the method based on Hellinger distance gives the best results.

However the originality and significance of this work are weak.

---

> ### Author Response · Authors · 2018-11-22
> **Response**
>
> Thanks for your comments.
> About our contributions, we want to make the following clarification,
>
> 1. This paper is more than simply comparing several training criteria in seq2seq models. Actually, from an intuitive observation and careful consideration, for the first time we categorize existing work into two categories only in terms of the loss function mathematics, and discover useful connections among those criteria (RAML and MRT are just different directions of the KL divergence/cross entropy, and both minimum divergence and maximum margin try to predict the evaluation score using $\log p$ during training), which is also the first insight for all these existing work from such a perspective. Much analysis (including the link between MRT and REINFORCE and the similarity and difference between minimum divergence and maximum margin) is original.
>
> 2. We propose a new training criterion based on the analysis of existing work, and the new training criterion improves the baseline by a large margin. The use of Hellinger distance is novel for sure.

---

### Meta-Review · Area_Chair1 · 2018-12-14
**Accept**

**Confidence:** 4
**Recommendation:** Accept (Poster)

**Metareview:**

The reviewers agree  that the paper is worthy of publication at ICLR, hence I recommend accept.

Regarding section 4.3 of the submission and the claim that this paper presents the first insight for existing work from a divergence minimization perspective, as pointed out by R2, I went and checked the details of RAML and they have similar insights in their equations (5) and (8). Please make this clearer in the paper. Regarding evaluation using greedy search instead of beam search, please consider using beam search for reporting test performance as this is the standard setup in sequence prediction. Please take my comments and the reviews into account an prepare the final version.